# Online Reciprocal Recommendation with Theoretical Performance Guarantees

**Fabio Vitale**
Department of Computer Science
Sapienza University of Rome (Italy) & University of Lille (France) & INRIA Lille Nord Europe
Rome, Italy & Lille, France
`fabio.vitale@inria.fr`

**Nikos Parotsidis**
University of Rome Tor Vergata, Rome, Italy
`nikos.parotsidis@uniroma2.it`

**Claudio Gentile**
INRIA Lille & Google New York
Lille, France & New York, USA
`cla.gentile@gmail.com`

## Abstract

A reciprocal recommendation problem is one where the goal of learning is not just to predict a user's preference towards a passive item (e.g., a book), but to recommend the targeted user on one side another user from the other side such that a mutual interest between the two exists. The problem thus is sharply different from the more traditional items-to-users recommendation, since a good match requires meeting the preferences at both sides. We initiate a rigorous theoretical investigation of the reciprocal recommendation task in a specific framework of sequential learning. We point out general limitations, formulate reasonable assumptions enabling effective learning and, under these assumptions, we design and analyze a computationally efficient algorithm that uncovers mutual likes at a pace comparable to that achieved by a clairvoyant algorithm knowing all user preferences in advance. Finally, we validate our algorithm against synthetic and real-world datasets, showing improved empirical performance over simple baselines.

## 1 Introduction

Recommendation Systems are at the core of many successful online businesses, from e-commerce, to online streaming, to computational advertising, and beyond. These systems have extensively been investigated by both academic and industrial researchers by following the standard paradigm of items-to-users preference prediction/recommendation. In this standard paradigm, a targeted user is presented with a list of items that s/he may prefer according to a preference profile that the system has learned based on both explicit user features (item data, demographic data, explicitly declared preferences, etc.) and past user activity. In more recent years, due to their hugely increasing interest in the online dating and the job recommendation domains, a special kind of recommendation systems called *Reciprocal Recommendation Systems* (RRS) have gained big momentum. The reciprocal recommendation problem is sharply different from the more traditional items-to-users recommendation, *since recommendations must satisfy both parties*, i.e., both parties can express their likes and dislikes and a good match requires meeting the preferences of both. Examples of RRS include, for instance: online recruitment systems (e.g., *LinkedIn*), [1] where a job seeker searches for jobs matching his/her preferences, say salary and expectations, and a recruiter who seeks suitable candidates to fulfil the job requirements; heterosexual online dating systems (e.g., *Tinder*), [2] where people have the common goal of finding a partner of the opposite gender; roommate matching systems

(e.g., *Badi*), [3] used to connect people looking for a room to those looking for a roommate, online mentoring systems, customer-to-customer marketplaces, etc.

From a Machine Learning perspective, the main challenge in a RRS is thus to learn *reciprocated* preferences, since the goal of the system is not just to predict a user's preference towards a passive item (a book, a movie, etc), but to recommend the targeted user on one side another user from the other side such that a mutual interest exists. Importantly enough, the interaction the two involved users have with the system is often *staged* and *unsynced*. Consider, for instance, a scenario where a user, Geena, is recommended to another user, Bob. The recommendation is successful only if both Geena and Bob mutually agree that the recommendation is good. In the first stage, Bob logs into the system and Geena gets recommended to him; this is like in a standard recommendation system: Bob will give a feedback (say, positive) to the system regarding Geena. Geena may never know that she has been recommended to Bob. In a subsequent stage, some time in the future, also Geena logs in. In an attempt to find a match, the system now recommends Bob to Geena. It is only when also Geena responds positively that the reciprocal recommendation becomes successful.

The problem of reciprocal recommendation has so far being studied mainly in the Data Mining, Recommendation Systems, and Social Network Analysis literature (e.g., [7, 1, 16, 15, 11, 19, 23, 3, 17, 12, 13]), with some interesting adaptations of standard collaborative filtering approaches to user feature similarity, but it has remained largely unexplored from a theoretical standpoint. Despite each application domain has its own specificity,[4] in this paper we abstract such details away, and focus on the broad problem of building matches between the two parties in the reciprocal recommendation problem based on behavioral information only. In particular, we do not consider *explicit* user preferences (e.g., those evinced by user profiles), but only the *implicit* ones, i.e., those derived from past user behavior. The explicit-vs-implicit user features is a standard dichotomy in Recommendation System practice, and it is by now common knowledge that collaborative effects (aka, implicit features) carry far more information about actual user preferences than explicit features, like, for instance, demographic metadata[18]. Similar experimental findings are also reported in the context of RRS in the online dating domain [2].

In this paper, we initiate a rigorous theoretical investigation of the reciprocal recommendation problem, and we view it as a sequential learning problem where learning proceeds in a sequence of rounds. At each round, a user from one of the two parties becomes active and, based on past feedback, the learning algorithm (called *matchmaker*) is compelled to recommend one user from the other party. The broad goal of the algorithm is to uncover as many mutual interests (called *matches*) as possible, and to do so *as quickly as possible*. We formalize our learning model in Section 2. After observing that, in the absence of structural assumptions about matches, learning is virtually precluded (Section 3), we come to consider a reasonable clusterability assumption on the preference of users at both sides. Under these assumptions, we design and analyze a computationally efficient matchmaking algorithm that leverages the correlation across matches. We show that the number of uncovered matches within $T$ rounds is comparable (up to constant factors) to those achieved by an optimal algorithm that knows beforehand all user preferences, provided $T$ and the total number of matches to be uncovered is not too small (Sections 3, and 4). Finally, in Section 5 we present a suite of initial experiments, where we contrast (a version of) our algorithm to noncluster-based random baselines on both synthetic and publicly available real-world benchmarks in the domain of online dating. Our experiments serve the twofold purpose of validating our structural assumptions on user preferences against real data, and showing the improved matchmaking performance of our algorithm, as compared to simple noncluster-based baselines.

## 2 Preliminaries

We first introduce our basic notation. We have a set of users $V$ partitioned into two parties. Though a number of alternative metaphores could be adopted here, for concreteness, we call the two parties $B$ (for "boys") and $G$ (for "girls"). Throughout this paper, $g$, $g'$ and $g''$ will be used to denote generic members of $G$, and $b$, $b'$, and $b''$ to denote generic members of $B$. For simplicity, we assume the two parties $B$ and $G$ have the same size $n$. A hidden ground truth about the mutual preferences of the members of the two parties is encoded by a sign function $\sigma : (B \times G) \cup (G \times B) \to \{-1, +1\}$.

Specifically, for a pairing $(b, g) \in B \times G$, the assignment $\sigma(b, g) = +1$ means that boy $b$ likes girl $g$, and $\sigma(b, g) = -1$ means that boy $b$ dislikes girl $g$. Likewise, given pairing $(g, b) \in G \times B$, we have $\sigma(g, b) = +1$ when girl $g$ likes boy $b$, and $\sigma(g, b) = -1$ when girl $g$ dislikes boy $b$. The ground truth $\sigma$ therefore defines a directed bipartite signed graph collectively denoted as $(\langle B, G \rangle, E, \sigma)$, where $E$, the set of directed edges in this graph, is simply $(B \times G) \cup (G \times B)$, i.e., the set of all possible $2n^2$ directed edges in this bipartite graph. A "+1" edge will sometimes be called a positive edge, while a "-1" edge will be called a negative edge. Any pair of directed edges $(g, b) \in G \times B$ and $(b, g) \in B \times G$ involving the same two subjects $g$ and $b$ is called a *reciprocal* pair of edges. We also say that $(g, b)$ is reciprocal to $(b, g)$, and vice versa. The pairing of signed edges $(g, b)$ and $(b, g)$ is called a *match* if and only if $\sigma(b, g) = \sigma(g, b) = +1$. The total number of matches will often be denoted by $M$. See Figure 1 for a pictorial illustration.

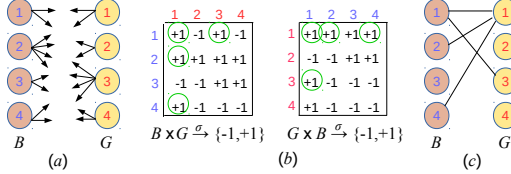

**Figure 1: (a)** The (complete and directed) bipartite graph $(\langle B, G \rangle, E, \sigma)$ with $n = |B| = |G| = 4$, edges are only sketched. **(b)** Representation of the $\sigma$ function through its two pieces $\sigma : B \times G \to \{-1, +1\}$ ($B \times G$ matrix on the left), and $\sigma : G \times B \to \{-1, +1\}$ ($G \times B$ matrix on the right). For instance, in this graph, Boy 1 likes Girl 1 and Girl 3, and dislikes Girl 2 and Girl 4, while Girl 3 likes Boy 1, and dislikes Boys 2, 3, and 4. Out of the $n^2 = 16$ pairs of reciprocal edges, this graph admits only $M = 4$ matches, which are denoted by green circles on both matrices. For instance, the pairing of edges $(1, 3)$ and $(3, 1)$ are a match since Boy 1 likes Girl 3 and, at the same time, Girl 3 likes Boy 1. **(c)** The associated (undirected and bipartite) matching graph $\mathcal{M}$. We have, for instance, $\deg_{\mathcal{M}}(\text{Girl } 1) = 3$, and $\deg_{\mathcal{M}}(\text{Boy } 2) = 1$.

Coarsely speaking, the goal of a learning algorithm $A$ is to uncover in a sequential fashion as many matches as possible as quickly as possible. More precisely, we are given a time horizon $T \leq n^2$, e.g., $T = n\sqrt{n}$, and at each round $t = 1, \ldots, T$:

($1_B$) $A$ receives the id of a boy $b$ chosen uniformly at random[5] from $B$ ($b$ is meant to be the "next boy" that logs into the system);

($2_B$) $A$ selects a girl $g' \in G$ to recommend to $b$;

($3_B$) $b$ provides feedback to the learner, in that the sign $\sigma(b, g')$ of the selected boy-to-girl edge is revealed to $A$.

Within the same round $t$, the three steps described above are subsequently executed after switching the roles of $G$ and $B$ (and will therefore be called Steps ($1_G$), ($2_G$), and ($3_G$)). Hence, each round $t$ is made up of two halves, the first half where a boy at random is logged into the system and the learner $A$ is compelled to select a girl, and the second half where a girl at random is logged in and $A$ has to select a boy. Thus at each round $t$, $A$ observes the sign of the two directed edges $(b, g')$ and $(g, b')$, where $b \in B$ and $g \in G$ are generated uniformly at random by the environment, and $g'$ and $b'$ are the outcome of $A$'s recommendation effort. Notice that we assume the ground truth encoded by $\sigma$ is persistent and noiseless, so that whereas the same user (boy or girl) may recur several times throughout the rounds due to their random generation, there is no point for the learner to request the sign of the same edge twice at two different rounds. The goal of algorithm $A$ is to maximize the number of uncovered matches within the $T$ rounds. The sign of the two reciprocal edges giving rise to a match need not be selected by $A$ in the same round; the round where the match is uncovered is the time when the reciprocating edge is selected, e.g., if in round $t_1$ we observe $\sigma(b, g') = -1$, $\sigma(g, b') = +1$, and in round $t_2 > t_1$ we observe $\sigma(b', g) = +1$, $\sigma(g'', b'') = +1$, we say that the match involving $b'$ and $g$ has been uncovered only in round $t_2$. In fact, if $A$ has uncovered a positive edge $g \to b'$ in (the second half of) round $t_1$, the reciprocating positive edge $(b', g)$ need not be uncovered any time soon, since $A$ has at the very least to wait until $b'$ will log into the system, an event which on average will occur only $n$ rounds later.

We call *matching graph*, and denote it by $\mathcal{M}$, the bipartite and *undirected* graph having $B \cup G$ as nodes, where $(b, g) \in B \times G$ is an edge in $\mathcal{M}$ if and only if $b$ and $g$ determine a match in the original graph $(\langle B, G \rangle, E, \sigma)$. Given $b \in B$, we let $\mathcal{N}_{\mathcal{M}}(b) \subseteq G$ be the set of matching girls for $b$ according to $\sigma$, and $\deg_{\mathcal{M}}(b)$ be the number of such girls. $\mathcal{N}_{\mathcal{M}}(g)$ and $\deg_{\mathcal{M}}(g)$ are defined symmetrically. See again Figure 1 for an example.

The performance of algorithm $A$ is measured by the number of matches found by $A$ within the $T$ rounds. Specifically, if $M_t(A)$ is the number of matches uncovered by $A$ after $t$ rounds of a given run,

we would like to obtain *lower* bounds on $M_T(A)$ that hold *with high probability* over the random generation of boys and girls that log into the system as well as the internal randomization of $A$. To this effect, we shall repeatedly use in our statements the acronym *w.h.p* to signify with probability at least $1 - \mathcal{O}(\frac{1}{n})$, as $n \to \infty$. It will also be convenient to denote by $E_t(A)$ the set of directed edges selected by $A$ during the first $t$ rounds, with $E_0(A) = \emptyset$. A given run of $A$ may therefore be summarized by the sequence $\{E_t(A)\}_{t=1}^T$. Likewise, $E_t^r(A)$ will denote the set of reciprocal (not necessarily matching) directed edges selected by $A$ up to time $t$. Finally, $E^r$ will denote the set of *all* $|B| \cdot |G| = n^2$ pairs of reciprocal (not necessarily matching) edges between $B$ and $G$.

We will first show (Section 3) that in the absence of further assumptions on the way the matches are located, there is not much one can do but try and simulate a random sampler. In order to further illustrate our model, the same section introduces a reference optimal behavior that assumes prior knowledge of the whole sign fuction $\sigma$. This will be taken as a yardstick to be contrasted to the performance of our algorithm SMILE (Section 4) that works under more specific, yet reasonable, structural assumptions on $\sigma$.

## 3 General limitations and optimal behavior

We now show[6] that in the absence of specific assumptions on $\sigma$, the best thing to do in order to uncover matches is to reciprocate at random, no matter how big the number $M$ of matches actually is.

**Theorem 1** *Given $B$ and $G$ such that $|B| = |G| = n$, and any integer $m \leq \frac{n^2}{2}$, there exists a randomized strategy for generating $\sigma$ such that $M = m$, and the expected number of matches uncovered by* any *algorithm $A$ operating on $(\langle B, G \rangle, E, \sigma)$ satisfies[7] $\mathbb{E}M_T(A) = \mathcal{O}\left(\frac{T}{n^2} M\right)$.*

An algorithm matching the above upper bound is described next. We call this algorithm OOMM (Oblivious Online Match Maker), The main idea is to develop a strategy that is able to draw uniformly at random as many pairs of reciprocal edges as possible from $E_r$ (recall that $E_r$ is the set of *all* reciprocal edges between $B$ and $G$). In particular, within the $T$ rounds, OOMM will draw uniformly at random $\Theta(T)$-many such pairs. The pseudocode of OOMM is given next. For brevity, throughout this paper an algorithm will be described only through Steps $(2_B)$ and $(2_G)$ – recall Section 2.

---
**Algorithm 1:** OOMM (Oblivious Online Match Maker)
---
▷ **INPUT** : $B$ and $G$

At each round $t$: $(2_B)$ Select $g'$ uniformly at random from $G$ ;
$\qquad\qquad (2_G)$ $B_{g,t} \leftarrow \{b'' \in B : (b'', g) \in E_t(\text{OOMM}), (g, b'') \notin E_{t-1}(\text{OOMM})\}$;
$\qquad\qquad\qquad$ **If** $B_{g,t} \neq \emptyset$ **then** select $b'$ uniformly at random from $B_{g,t}$
$\qquad\qquad\qquad\qquad$ **else** select $b'$ uniformly at random from $B$ .

---

OOMM simply operates as follows. In Step $(2_B)$ of round $t$, the algorithm chooses a girl $g'$ uniformly at random from the whole set $G$. OOMM maintains over time the set $B_{g,t} \subseteq B$ of all boys that so far gave their feedback (either positive or negative) on $g$, but for whom the feedback from $g$ is not available yet. In Step $(2_G)$, if $B_{g,t}$ is not empty, OOMM chooses a boy uniformly at random from $B_{g,t}$, otherwise it selects a boy uniformly at random from the whole set $B$.[8]

Note that, the way it is designed, the selection of $g'$ and $b'$ does *not* depend on the signs $\sigma(b, g)$ or $\sigma(g, b)$ collected so far. The following theorem guarantees that $\mathbb{E}M_T(\text{OOMM}) = \Theta\left(\frac{T}{n^2} M\right)$, which is as if we were able to directly sample in most of the $T$ rounds pairs of reciprocal edges.

**Theorem 2** *Given any input graph $(\langle B, G \rangle, E, \sigma)$, with $|B| = |G| = n$, if $T - n = \Omega(n)$ then $E_T^r(\text{OOMM})$ is selected uniformly at random (with replacement) from $E^r$, its size $|E_T^r(\text{OOMM})|$ is such that $\mathbb{E}|E_T^r(\text{OOMM})| = \Theta(T)$, and the expected number of matches disclosed by OOMM is such that $\mathbb{E}M_T(\text{OOMM}) = \Theta\left(\frac{T}{n^2} M\right)$.*

We now describe an optimal behavior (called *Omniscient Matchmaker*) that assumes prior knowledge of the whole edge sign assignment $\sigma$. This optimal behavior will be taken as a reference performance for our algorithm of Section 4. This will also help to better clarify our learning model.

**Definition 1** *The* Omniscient Matchmaker $A^*$ *is an* optimal *strategy based on the prior knowledge of the signs* $\sigma(b,g)$ *and* $\sigma(g,b)$ *for all* $b \in B$ *and* $g \in G$. *Specifically, based on this information,* $A^*$ *maximizes the number of matches uncovered during* $T$ *rounds over* all $n^{2T}$ *possible selections that can be made in Steps* $(2_B)$ *and* $(2_G)$. *We denote this optimal number of matches by* $M_T^* = M_T(A^*)$.

Observe that when the matching graph $\mathcal{M}$ is such that $\deg_{\mathcal{M}}(u) > \frac{T}{n}$ for some user $u \in B \cup G$, no algorithm will be able to uncover all $M$ matches in expectation, since Steps $(1_B)$ and $(1_G)$ of our learning protocol entail that the expected number of times each user $u$ logs into the system is equal to $\frac{T}{n}$. In fact, this holds *even* for the Omniscient Matchmaker $A^*$, despite the prior knowledge of $\sigma$. For instance, when $\mathcal{M}$ turns out to be a random bipartite graph[9] the expected number of matches that any algorithm can achieve is always upper bounded by $\mathcal{O}\left(\frac{T}{n^2}M\right)$ (this is how Theorem 1 is proven). On the other hand, in order to have $M_T^* = \Theta(M)$ as $n$ grows large, it is sufficient that $\deg_{\mathcal{M}}(u) \leq \frac{T}{n}$ holds for all users $u \in B \cup G$, even with such a random $\mathcal{M}$. In order to avoid the pitfalls of $\mathcal{M}$ being a random bipartite graph (and hence the negative result of Theorem 1), we need to slightly depart from our general model of Section 2, and make structural assumptions on the way matches can be generated. The next section formulates such assumptions, and analyzes an algorithm that under these assumptions is essentially optimal i.t.o. number of uncovered matches. The assumptions and the algorithm itself are then validated against simple baselines on real-world data in the domain of online dating (Section 5).

## 4  A model based on clusterability of received feedback

In a nutshell, our model is based on the extent to which it is possible to arrange the users in (possibly) *overlapping* clusters by means of the feedbacks they may potentially receive from the other party. In order to formally describe our model, it will be convenient to introduce the Boolean preference matrices $\boldsymbol{B}, \boldsymbol{G} \in \{0,1\}^{n \times n}$. These two matrices collect in their rows the ground truth contained in $\sigma$, separating the two parties $B$ and $G$. Specifically, $\boldsymbol{B}_{i,j} = \frac{1}{2}(1 + \sigma(b_i, g_j))$, and $\boldsymbol{G}_{i,j} = \frac{1}{2}(1 + \sigma(g_i, b_j))$ (these are essentially the matrices exemplified in Figure 1(b) where the "$-1$" signs therein are replaced by "0"). Then, we consider the $n$ column vectors of $\boldsymbol{B}$ (resp. $\boldsymbol{G}$) – i.e., the whole set of feedbacks that each $g \in G$ (resp. $b \in B$) may receive from members of $B$ (resp. $G$) and, for a given radius $\rho \geq 0$, the associated covering number of this set of Boolean vectors w.r.t. Hamming distance. We recall that the covering number at radius $\rho$ is the smallest number of balls of radius $\leq \rho$ that are needed to cover the entire set of $n$ vectors. The smaller $\rho$ the higher the covering number. If the covering number stays small despite a small $\rho$, then our $n$ vectors can be clustered into a small number of clusters each one having a small (Hamming) radius.

As we mentioned in Section 3, a reasonable model for this problem is one for which our learning task can be solved in a nontrival manner, thereby specifically avoiding the pitfalls of $\mathcal{M}$ being a random bipartite graph. It is therefore worth exploring what pairs of radii and covering numbers may be associated with the two preference matrices $\boldsymbol{G}$ and $\boldsymbol{B}$ when $\mathcal{M}$ is indeed random bipartite. Assume $M = o(n^2)$, so as to avoid pathological cases. When $\mathcal{M}$ is random bipartite, one can show that we may have $\rho = \Omega\left(\frac{M}{n}\right)$ even when the two covering numbers are both 1. Hence, the only interesting regime is when $\rho = o\left(\frac{M}{n}\right)$. Within this regime, our broad modeling assumption is that the resulting covering numbers for $\boldsymbol{G}$ and $\boldsymbol{B}$ are $o(n)$, i.e., less that linear in $n$ when $n$ grows large.

**Related work.** The approach of clustering users according to their description/preference similarities while exploiting user feedback is similar in spirit to the two-sided clusterability assumptions investigated, e.g., in [1], which is based on a mixture of explicit and implicit (collaborative filtering-like) user features. Yet, as far as we are aware, ours is the first model that lends itself to a rigorous theoretical quantification of matchmaking performance (see Section 4.1). Moreover, in general in our case the user set is not partitioned as in previous RRS models. Each user may in fact belong to more than one cluster, which is apparently more natural for this problem.

The reader might also wonder whether the reciprocal recommendation task and associated modeling assumptions share any similarity to the problem of (online) matrix completion/prediction. Recovering a matrix from a sample of its entries has been widely analyzed by a number of authors with different approaches, viewpoints, and assumptions, e.g., in Statistics and Optimization (e.g., [5, 14]), in Online Learning (e.g., [20, 21, 22, 9, 8, 6, 10]), and beyond. In fact, one may wonder if the problem of predicting the entries of matrices $\boldsymbol{B}$ and $\boldsymbol{G}$ may somehow be equivalent to the problem of disclosing

matches between $B$ and $G$. A closer look reveals that the two tasks are somewhat related, but not quite equivalent, since in reciprocal recommendation the task is to search for matching "ones" between the two binary matrices $\boldsymbol{B}$ and $\boldsymbol{G}$ by observing entries of the two matrices *separately*. In addition, because we get to see at each round the sign of two pairings $(b, g')$ and $(g, b')$, where $b$ and $g$ are drawn at random and $b'$ and $g'$ are selected by the matchmaker, our learning protocol is rather half-stochastic and half-active, which makes the way we gather information about matrix entries quite different from what is usually assumed in the available literature on matrix completion.

## 4.1 An efficient algorithm

Under the above modeling assumptions, our goal is to design an efficient matchmaker. We specifically focus on the ability of our algorithm to disclose $\Theta(M)$ matches, in the regime where also the optimal number of matches $M_T^*$ is $\Theta(M)$. Recall from Section 3 that the latter assumption is needed so as to make the uncovering of $\Theta(M)$ matches possible within the $T$ rounds. Our algorithm, called SMILE (Sampling Matching Information Leaving out Exceptions) is described as Algorithm 2. The algorithm depends on input parameter $S \in [\log n, n/\log n]$ and, after randomly shuffling both $B$ and $G$, operates in three phases: Phase 0 (described at the end), Phase I, and Phase II.

---

**Algorithm 2:** SMILE (**S**ampling **M**atching **I**nformation **L**eaving out **E**xceptions)

▷ **INPUT** : $B$ and $G$; parameter $S > 0$.
Randomly shuffle sets $B$ and $G$ ;
Phase 0: Run OOMM to provide an estimate $\hat{M}$ of $M$ ;
Phase I: $(\mathcal{C}, \mathcal{F}) \leftarrow$ *Cluster Estimation*$(\langle B, G\rangle, S)$;
Phase II: *User Matching*$(\langle B, G\rangle, (\mathcal{C}, \mathcal{F}))$;

---

Due to space limitations, the actual pseudocode of Cluster Estimation() and User Matching() is presented in the appendix. What follows is an an informal, yet precise, description of their functioning.

**Phase I: Cluster Estimation.** SMILE approximates the clustering over users by: i. asking, for each cluster representative $b \in B$, $\Theta(n)$ feedbacks (i.e., edge signs) selected at random from $G$ (and operating symmetrically for each representative $g \in G$), ii. asking $\Theta(S)$-many feedbacks for each remaining user, where parameter $S$ will be set later. In doing so, SMILE will be in a position to estimate the clusters each user belongs to, that is, to estimate the matching graph $\mathcal{M}$, the misprediction per user being w.h.p of the order of $(n \log n)/S$. The estimated $\mathcal{M}$ will then be used in Phase II.

A more detailed description of the Cluster Estimation procedure follows. For convenience, we focus on clustering $G$ (hence observing feedbacks from $B$ to $G$), the procedure operates in a completely symmetric way on $B$. Let $F_g$ be the set of all $b \in B$ who provided feedback on $g \in G$ so far. Assume for the moment we have at our disposal a subset $G^r \subseteq G$ containing one representative for each cluster over $B$, and that for each $g \in G^r$ we have already observed $n/2$ feedbacks provided by $n/2$ *distinct* members of $B$, selected *uniformly at random* from $B$. Also, let $B(g, S)$ be a subset of $B$ obtained by sampling at random $S' = 2S + 4\sqrt{S \log n}$-many $b$ from $B$. Then a Chernoff-Hoeffding bound argument shows that for any $g \in G \setminus G^r$ and any $g^r \in G^r$ we have w.h.p. $|B(g, S) \cap F_{g^r}| \geq S$. We use the above to estimate the cluster each $g \in G \setminus G^r$ belongs to. This task can be accomplished by finding $g^r \in G^r$ who receives the same set of feedbacks as that of $g$, i.e., who belongs to the same cluster as $g^r$. Yet, in the absence of the feedback provided by *all* $b \in B$ to both $g$ and $g^r$, it is not possible to obtain this information with certainty. The algorithm simply *estimates* $g$'s cluster by exploiting Step $(1_B)$ of the protocol to ask for feedback on $g$ from $S' = S'(S)$ randomly selected $b \in B$, which will be seen as forming the subset $B(g, S)$. We shall therefore assign $g$ to the cluster represented by an arbitrary $g^r \in G^r$ such that $s(b, g) = s(b, g^r)$ for all $b \in B(g, S) \cap F_{g^r}$. We proceed this way for all $g \in G \setminus G^r$.

We now remove the assumption on $G^r$. Although we initially do not have $G^r$, we can build through a concentration argument an approximate version of $G^r$ while asking for the feedback $B(g, S)$ on each unclustered $g$. The Cluster Estimation procedure does so by processing girls $g$ sequentially, as described next. Recall that $G$ was randomly shuffled into an ordered sequence $G = \{g_1, g_2, \ldots, g_n\}$. The algorithm maintains an index $i$ over $G$ that only moves forward, and collects feedback information for $g_i$. At any given round, $G^r$ contains all cluster representatives found so far. Given $b \in B$ that needs to be served during round $t$ (Step $(1_B)$), we include $b$ in $F_{g_i}$. If $|F_{g_i}|$ becomes as big as $S'$, then we look for $g \in G^r$ so as to estimate $g_i$'s cluster. If we succeed, index $i$ is incremented and the algorithm will collect feedback for $g_i$ during the next rounds. If we do not succeed, $g_i$ will be included in $G^r$, and the algorithm will continue to collect feedback on $g_i$ until $|F_{g_i}| < \frac{n}{2}$. When

$|F_{g_i}| \geq \frac{n}{2}$, index $i$ is incremented, so as to consider the next member of $G$. Phase I terminates when we have estimated the cluster of each $b$ and $g$ that are themselves not representative of any cluster.

Finally, when we have concluded with one of the two sides, but not with the other (e.g., we are done with $G$ but not with $B$), we continue with the unterminated side, while for the terminated one we can select members ($g \in G$ in this case) in Step 2 (Step ($2_B$) in this case) arbitrarily.

**Phase II: User matching.** In phase II, we exploit the feedback collected in Phase I so as to match as many pairs $(b, g)$ as possible. For each user $u \in B \cup G$ selected in Step ($1_B$) or Step ($1_G$), we pick in step ($2_G$) or ($2_B$) a user $u'$ from the other side such that $u'$ belongs to an estimated cluster which is among the set of clusters whose members are liked by $u$, and viceversa. When no such $u'$ exists, we select $u'$ from the other side arbitrarily.

**Phase 0: Estimating $M$.** In the appendix we show that the optimal tuning of $S$ is to set it as a function of the number of hidden matches $M$, i.e. $S := (n^2 \log n)/M$. Since $M$ is unknown, we run a preliminary phase where we run OOMM (from Section 3) for a few rounds. Using Theorem 2 it is not hard to show that the number $T_{\hat{M}}$ of rounds taken by this preliminary phase to find an estimate $\hat{M}$ of $M$ which is w.h.p. accurate up to a constant factor satisfies $T_{\hat{M}} = \Theta\left((n^2 \log n)/M\right)$.

In order to quantify the performance of SMILE, it will be convenient to refer to the definition of the Boolean preference matrices $\boldsymbol{B}, \boldsymbol{G} \in \{0,1\}^{n \times n}$. For a given radius $\rho \geq 0$, we denote by $C_\rho^G$ the covering number of the $n$ column vectors of $\boldsymbol{B}$ w.r.t. Hamming distance. In a similar fashion we define $C_\rho^B$. Moreover, let $C^G$ and $C^B$ be the total number of cluster representatives for girls and boys, respectively, found by SMILE, i.e., $C^G = |G^r|$ and $C^B = |B^r|$ at the end of the $T$ rounds. The following theorem shows that when the optimal number of matches $M_T^*$ is $M$, then so is also $M_T(\text{SMILE})$ up to a constant factor, provided $M$ and $T$ are not too small.

**Theorem 3** *Given any input graph $(\langle B, G \rangle, E, \sigma)$, with $|B| = |G| = n$, such that $M_T^* = M$ w.h.p. as $n$ grows large, then we have*

$$C^G \leq \bar{C}^G \stackrel{\text{def}}{=} \min\left\{\min_{\rho \geq 0}\left(C_{\rho/2}^G + 3\rho S'\right), n\right\},$$
$$C^B \leq \bar{C}^B \stackrel{\text{def}}{=} \min\left\{\min_{\rho \geq 0}\left(C_{\rho/2}^B + 3\rho S'\right), n\right\}.$$

*Furthermore, when $T$ and $M$ are such that $T = \omega\left(n(\bar{C}^G + \bar{C}^B + S')\right)$ and $M = \omega\left(\frac{n^2 \log n}{S}\right)$, then we have w.h.p. $M_T(\text{SMILE}) = \Theta(M)$.*

Notice in the above theorem the role played by the upper bounds $\bar{C}^G$ and $\bar{C}^B$. If the minimizing $\rho$ therein gives $\bar{C}^G = \bar{C}^B = n$, we have enough degrees of freedom for $\mathcal{M}$ to be generated as a random bipartite graph. On the other hand, when $\bar{C}^G$ and $\bar{C}^B$ are significantly smaller than $n$ at the minimizing $\rho$ (which is what we expect to happen in practice) the resulting $\mathcal{M}$ will have a cluster structure that cannot be compatible with a random bipartite graph. This entails that on both sides of the bipartite graph, each subject receives from the other side a set of preferences that can be collectively clustered into a relatively small number of clusters with small intercluster distance. Then the number of rounds $T$ that SMILE takes to achieve (up to a constant factor) the same number of matches $M_T^*$ as the Omniscient Matchmaker drops significantly. In particular, when $S$ in SMILE has the form $(n^2 \log n)/\hat{M}$, where $\hat{M}$ is the value returned by Phase 0, we have the following result.

**Corollary 1** *Given any input graph $(\langle B, G \rangle, E, \sigma)$, with $|B| = |G| = n$, such that $M_T^* = M$ w.h.p. as $n$ grows large, with $T$ and $M$ satisfying $T = \omega\left(n\left(\bar{C}^G + \bar{C}^B\right) + (n^3 \log n)/M\right)$, where $\bar{C}^G$ and $\bar{C}^B$ are the upper bounds on $C_G$ and $C_B$ given in Theorem 3, then we have w.h.p. $M_T(\text{SMILE}) = \Theta(M)$.*

In order to evaluate in detail the performance of SMILE, it is very interesting to show to what extent the conditions bounding from below $T$ in Theorem 3 are necessary. We have the following general limitation, holding for any matchmaker $A$.

**Theorem 4** *Given $B$ and $G$ such that $|B| = |G| = n$, any integer $m \in (n \log n, n^2 - n \log n)$, and any algorithm $A$ operating on $(\langle B, G \rangle, E, \sigma)$, there exists a randomized strategy for generating $\sigma$ such that $m - \frac{n}{C_0^G + C_0^B - 1} < M \leq m$, and the number of rounds $T$ needed to achieve $\mathbb{E}M_T(A) = \Theta(M)$, satisfies $T = \Omega(n(C_0^G + C_0^B) + M)$, as $n \to \infty$.*

**Remark 1** *One can verify that the time bound for SMILE established in Corollary 1 is nearly optimal whenever $M = \omega\left(n^{3/2}\sqrt{\log n}\right)$. To see this, observe that by definition we have $\bar{C}^G \leq C_0^G$ and*

$\bar{C}^B \leq C_0^B$. Now, if $M = \omega\left(n^{3/2}\sqrt{\log n}\right)$, then the additive term $(n^3 \log n)/M$ becomes $o(M)$ and the condition on $T$ in Corollary 1 simply becomes $T = \omega\left(n\left(C_0^G + C_0^B + M'\right)\right)$, where $M' = o(M)$. This has to be contrasted to the lower bound on $T$ contained in Theorem 4.

We now explain why it is possible that, when $M = \omega\left(n^{3/2}\sqrt{\log n}\right)$, the additive term $(n^3 \log n)/M$ in the bound $T = \omega\left(n\left(\bar{C}^G + \bar{C}^B\right) + (n^3 \log n)/M\right)$ of Corollary 1 becomes $o(M)$, while the first term $n\left(\bar{C}^G + \bar{C}^B\right)$ can be upper bounded by $n\left(C_0^G + C_0^B\right)$. Since the lower bound $T = \Omega(n\left(C_0^G + C_0^B\right) + M)$ of Theorem 4 has a linear dependence on $M$, it might seem surprising that the larger $M$ is the smaller becomes the second term in the bound of Corollary 1. However, it is important to take into account that $T$ in Corollary 1 must be large enough to also satisfy the condition $M_T^* = M$. Let $T^*$ be the number of rounds $T$ necessary to satisfy w.h.p. $M_T^* = M$. In Corollary 1, both the conditions $T \geq T^*$ and $T = \omega\left(n\left(\bar{C}^G + \bar{C}^B\right) + (n^3 \log n)/M\right)$ must simultaneously hold. When $M$ is large, the number of rounds needed to satisfy the former condition becomes much larger than the one needed for the latter.

As a further insight, consider the following. We either have $M = \mathcal{O}\left(n(\bar{C}^G + \bar{C}^B)\right)$ or $M = \omega\left(n(\bar{C}^G + \bar{C}^B)\right)$. In the first case, the lower bound in Theorem 4 clearly becomes $T = \Omega\left(n\left(C_0^G + C_0^B + \bar{C}^G + \bar{C}^B\right)\right)$, hence not directly depending on $M$. In the second case, whenever $M = \omega\left(n^{3/2}\sqrt{\log n}\right)$, $T^*$ is larger than $n\left(\bar{C}^G + \bar{C}^B\right) + (n^3 \log n)/M$ since, by definition, we must have $T^* = \Omega(M)$, while in this case $n\left(\bar{C}^G + \bar{C}^B\right) + (n^3 \log n)/M = o(M)$. In conclusion, if the number of rounds SMILE takes to uncover $\Theta(M)$ matches equals the number of rounds taken by the omniscient Matchmaker to uncover exactly $M$ matches, then SMILE is optimal up to a constant factor, because no algorithm can outperform the omniscient Matchmaker. This provides a crucially important insight into the key factors allowing the additive term $(n^3 \log n)/M$ to be equal to $o(M)$ in Corollary 1, and is indeed one of the keystones in the proof of Theorem 3.

We conclude this section by emphasizing the fact that SMILE is indeed quite scalable. As proven in the appendix, an implementation of SMILE exists that leverages a combined use of suitable data-structures, leading to both time and space efficiency.

**Theorem 5** Let $\bar{C}^G$ and $\bar{C}^B$ be the upper bounds on $C_G$ and $C_B$ given in Theorem 3. Then the running time of SMILE is $\mathcal{O}\left(T + n\,S\left(\bar{C}^G + \bar{C}^B\right)\right)$, the memory requirement is $\mathcal{O}(n\,(\bar{C}^G + \bar{C}^B))$. Furthermore, when $T = \omega\left(n\left(\bar{C}^G + \bar{C}^B\right) + (n^3 \log n)/M\right)$, as required by Corollary 1, the amortized time per round is $\Theta(1) + o(\bar{C}^G + \bar{C}^B)$, which is always sublinear in $n$.

## 5  Experiments

In this section, we evaluate the performance of (a variant of) our algorithm by empirically contrasting it to simple baselines against artificial and real-world datasets from the online dating domain. The comparison on real-world data also serves as a validation of our modeling assumptions.

| | | | | | #clusters within bounded radius | | | | | |
|---|---|---|---|---|---|---|---|---|---|---|
| | | properties | | | $2 \cdot n/\log n$ | | $n/\log n$ | | $0.5 \cdot n/\log n$ | |
| Synthetic datasets (2000 boys and 2000 girls) | | | | | | | | | | |
| | $C_B$ | $C_G$ | #likes | #matches | $C_B$ | $C_G$ | $C_B$ | $C_G$ | $C_B$ | $C_G$ |
| S-20-23 | 20 | 23 | $2.45M$ | $374K$ | 20 | 23 | 20 | 23 | 445 | 429 |
| S-95-100 | 95 | 100 | $2.46M$ | $377K$ | 95 | 100 | 95 | 100 | 603 | 624 |
| S-500-480 | 500 | 480 | $2.47M$ | $380K$ | 500 | 480 | 500 | 480 | 983 | 950 |
| S-2000-2000 | 2000 | 2000 | $2.47M$ | $382K$ | 2000 | 2000 | 2000 | 2000 | 2000 | 2000 |
| Real-world datasets | | | | | | | | | | |
| | $|B|$ | $|G|$ | #likes | #matches | $C_B$ | $C_G$ | $C_B$ | $C_G$ | $C_B$ | $C_G$ |
| RW-1007-1286 | 1007 | 1286 | $125K$ | $13.9K$ | 53 | 48 | 177 | 216 | 385 | 508 |
| RW-1526-2564 | 1526 | 2564 | $227K$ | $19.6K$ | 37 | 45 | 138 | 216 | 339 | 601 |
| RW-2265-3939 | 2265 | 3939 | $370K$ | $25.0K$ | 42 | 45 | 145 | 215 | 306 | 622 |

**Table 1:** Relevant properties of our datasets. The last six columns present an approximation to the number of clusters when we allow radius $2 \cdot n/\log n$, $n/\log n$, and $0.5 \cdot n/\log n$ between users of the same cluster.

**Datasets.** The relevant properties of our datasets are given in Table 1. Each of our synthetic datasets has $|B| = |G| = 2000$. We randomly partitioned $B$ and $G$ into $C_B$ and $C_G$ clusters, respectively. Each boy likes all the girls of a cluster $C$ with probability $0.2$, and with probability $0.8$ dislikes them.

We do the same for the preferences from girls to boy clusters. Finally, for each preference (either positive or negative) we reverse its sign with probability $1/(2 \cdot \log n)$ (in our case, $n = 2000$). Notice that in Table 1, for all four datasets we generated, the number of likes is bigger than $|B| \cdot |G|/2$. As for real-world datasets, we used the one from [4], which is also publicly available. This is a dataset from a Czech dating website, where 220,970 users rate each other in a scale from 1 (worst) to 10 (best). The gender of the users is not always available. To get two disjoint parties $B$ and $G$, where each user rates only users from the other party, we disregarded all users whose gender is not specified. As this dataset is very sparse, we extracted dense subsets as follows. We considered as "like" any rating $> 2$, while all ratings, including the missing ones, are "dislikes". Next, we iteratively removed the users with the smallest number of ratings until we met some desired density level. Specifically, we executed the above process until we obtained two sets $B$ and $G$ such that the number of likes between the two parties is at least $2(\min\{|B|,|G|\})^{3/2}$ (resulting in dataset RW-1007-1286), $1.75(\min\{|B|,|G|\})^{3/2}$ (dataset RW-1526-2564), or $1.5(\min\{|B|,|G|\})^{3/2}$ (dataset RW-2265-3939).

**Random baselines.** We included as baselines OOMM , from Section 3, and a random method that asks a user for his/her feedback on another user (of opposite gender) picked uniformly at random. We refer to this algorithm as UROMM.

**Implementation of SMILE.** In the implementation of SMILE, we slightly deviated from the description in Section 4.1. One important modification is that we interleaved Phase I and Phase II. The high-level idea is to start exploiting immediately the clusters once some clusters are identified, without waiting to learn all of them. Additionally, we gave higher priority to exploring the reciprocal feedback of a discovered like, and we avoided doing so in the case of a dislike. Finally, whenever we test whether two users belong in the same cluster, we allowed a radius of a $(1/\log n)$ fraction of the tested entries. The parameter $S'$ in SMILE has been set to $S + \sqrt{S \log n}$, with $S = (n^2 \log n)/\hat{M}$, where $\hat{M}$ is the estimate from Phase 0. We call the resulting algorithm I-SMILE (Improved SMILE).

**Evaluation.** To get a complete picture on the behavior of the algorithms for different time horizons, we present for each algorithm the number of discovered matches as a function of $T \in \{1, \ldots, 2|B||G|\}$. Figure 2 contains three representative cases. In all datasets we tested, I-SMILE clearly outperforms UROMM and OOMM. Our experiments confirm that SMILE (and therefore I-SMILE) quickly learns the underlying structure of the likes between users, and uses this structure to reveal the matches between them. Moreover, the variant I-SMILE that we implemented allows one not only to perform well on graphs with no underlying structure in the likes, but also to discover matches during the exploration phase while learning the clusters.

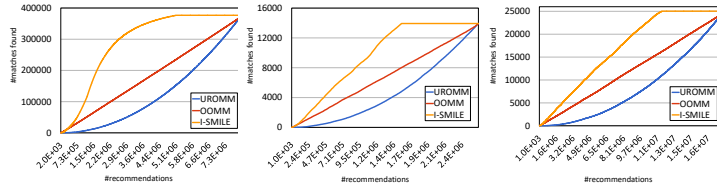

**Figure 2:** Empirical comparison of the three algorithms on datasets S-95-100 (left), RW-1007-1286 (middle), RW-2265-3939 (right). Each plot gives number of disclosed matches vs. time.

## 6 Conclusions and Ongoing Research

We have initiated a theoretical investigation of the problem of reciprocal recommendation in an ad hoc model of sequential learning. Under suitable clusterability assumptions, we have introduced an efficient matchmaker called SMILE , and have proven its ability to uncover matches at a speed comparable to the omniscient Matchmaker, so long as $M$ and $T$ are not too small (Theorem 3 and Corollary 1). Our theoretical findings also include a computational complexity analysis (Theorem 5), as well as limitations on the number of disclosable matches in both the general (Theorem 1) and the cluster case (Theorem 4). We complemented our results with an initial set of experiments on synthetic and real-world datasets in the online dating domain, showing encouraging evidence.

Current ongoing research includes: i. Introducing suitable noise models for the sign function $\sigma$. ii. Generalizing our learning model to nonbinary feedback preferences. iii. Investigating algorithms whose goal is to maximize the area under the curve "number of matches-vs-time", i.e., the criterion $\sum_{t \in [T]} M_t(A)$ , rather than the one we analyzed in this paper; maximizing this criterion requires interleaving the phases where we collect matches (exploration) and the phases where we do actually disclose them (exploitation). iv. More experimental comparisons on different datasets against heuristic approaches available in the literature. v. Incorporating in the protocol different frequencies for the user logins.

**Acknowledgements**

We would like to thank the anonymous reviewers for their valuable comments and suggestions that helped improving the presentation of this paper. Special thanks to Flavio Chierichetti and Marc Tommasi for helpful discussions in early stages of our investigation. Fabio Vitale acknowledges support from the ERC Starting Grant "DMAP 680153", the Google Focused Award "ALL4AI", and grant "Dipartimenti di Eccellenza 2018-2022", awarded to the Department of Computer Science of Sapienza University.

## Footnotes

[1] `https://www.linkedin.com/`.

[2] `https://tinder.com`.

[3] https://badiapp.com/en.

[4] For instance, users in an online dating system have relevant visual features, and the system needs specific care in removing popular user bias, i.e., ensuring that popular users are not recommended more often than unpopular ones.

[5] Though different distributional assumptions could be made, for technical simplicity in this paper we decided to focus on the uniform distribution only.

[6] All proofs are provided in the appendix.

[7] Recall that an *upper* bound on $M_T(A)$ is a negative result here, since we are aimed at making $M_T(A)$ as large as possible.

[8] A boy could be selected more than once while serving a girl $g$ during the $T$ rounds. The optimality of OOMM (see Theorems 1 and 2) implies that this redundancy does not significantly affect OOMM's performance.

[9] The matching graph $\mathcal{M}$ is a random bipartite graph if any edge $(b, g) \in B \times G$ is generated independently with the same probability $p \in [0, 1]$.

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
