[Reviews · NeurIPS 2018]

Reviewer 1



This submission explores theoretical properties of the reciprocal recommendation task (both parties must agree to match), introduces an algorithm to make recommendations, and validates the algorithm with synthetic and real-world data. This is a very interesting problem, and quite distinct from the typical item recommendation task. It is certainly a task worth exploring in greater detail theoretically and practically. As far as I can tell, the theoretical analysis seems sound and very relevant for those interested in understanding the problem in greater detail. I would be interested in some exploration of any fairness implications of such a system for users with different behaviors. For example, if one user log into the system more frequently, how will their experience be different from a user that logs in less frequently? What about users that have fewer interactions in general? It would be interesting to explore issues of disproportionate impact within such a system where behavior properties of users lead them to receive different recommendation treatments. However, I understand that space is limited already, and that this might be best suited for future work. The writing is very good, and the authors maintain a reasonable narrative, which is difficult in theoretical papers. Thank you for making the effort to make your paper readable! The contributions of this paper are original and relevant to the real-world applications of such systems. This work has the potential to open new areas of theoretical analyses of these systems as well as to impact the deployment of algorithms on real-world platforms.

Reviewer 2



[Summary] This paper studies an online reciprocal recommendation problem. The authors first model the problem as the recommendation problem in which, at each time, boy b randomly chosen from set B visits a matching site and the site recommend girl g in set G to the boy and the same process is repeated by switching the roles of boys and girls. They assume a function \sigma from (G\times B)\cup(B\times G) to {-1,1}. The objective is maximization of the uncovered matched pairs by time T. They first proposed algorithm OOMM that recommends b/g to g/b for which \sigma(b,g)/\sigma(g,b) is known. Next they proposed algorithm SMILE that estimates clusters of (boys b)/(girls g) which have similar preference function \sigma(x,b)/\sigma(x,g), and recommend b/g to g/b for which b/g is a member of an estimated cluster whose \sigma(x,b)/\sigma(x,g) is likely to be 1 for (x=g)/(x=b). They proved that the number of the matched pairs uncovered by OOMM is proportional to the rate of known function values but that by SMILE is proportional to the number of the true matched pairs if the number of known function values are enough large. They also empirically demonstrated the superiority of their algorithms by comparing the number of matched pairs uncovered by them and with that by a random selection algorithm. [Originality] Their formalization, algorithms and theoretical and empirical results look new. [Strengths] * Clear formalization of the problem. * Theoretically guaranteed algorithms. * Performance superiority of their proposed algorithms for large real-world datasets compared to the performance of a random algorithm. [Weakness] * How to set the parameter S remains a problem. * Algorithm SMILE is interesting but their theoretical results on its performance is not easy to interpret. * No performance comparison with existing algorithms [Recommendation] I recommend this paper to be evaluated as "a good submission; an accept". Their problem formalization is clear, and SMILE algorithm and its theoretical results are interesting. All their analyses are asymptotically evaluated, so I worry about how large the constant factors are. It would make this manuscript more valuable if how good their algorithms (OOMM & SMILE) would be shown theoretically and empirically compared to other existing algorithms. [Detailed Comments] p.7 Th 3 & Cor 1: C^G and C^B look random variables. If it is true, then they should not be used as parameters of T's order. Maybe the authors want to use their upper bounds shown above instead of them. p.7 Sec. 5 : Write the values of parameter S. [Comments to Authors’ Feedback] Setting parameter S: Asymptotic relation shown in Th 3 is a relation between two functions. It is impossible to estimate S from estimated M for a specific n using such a asymptotic functional relation.

Reviewer 3



The authors consider the problem of online reciprocal recommendation (ORR), where a recommendation is successful if it satisfies both parties. The online reciprocal recommendation is of practical importance as it can model online systems for job-matching, online dating, roommate matching and so on. The authors model ORR as a planted bipartite graph where the two parties form the two partites and edges are hidden. A query is directional between partites and the system gets to place two queries u->v’ and v->u’ on each time slot, where u and v are randomly chosen by nature. Moreover, an edge uv is established iff both u->v and v->u queries are made. Under this model, the authors investigate how many such edges can be established in T = o(n^2) rounds. They present negative results when a planted graph is unrestricted. In particular, there are graphs where an oblivious algorithm is optimal. Whereas, when the feedbacks are clusterable they provide an efficient algorithm, namely SMILE, that learn an order-wise optimal number of edges. The paper also validates the theoretical guarantees by testing them on a real-world dataset. Pros: * The author initiates a theoretical study of the practically important problem of online reciprocal recommendation. * The proposed model of the ORR is novel and allows for key structural insights. The quantification of clustering using a notion of covering number is also interesting. * The author provides a negative result: without structural assumption learning is not possible. Specifically, an oblivious online matching strategy matches the best possible performance. * On the positive side the authors show that when the received feedback is clusterable then through joint cluster learning and user matching order-wise optimal learning is possible, i.e. the number of edges uncovered by SMILE is of the same order of an algorithm that knows the planted bipartite graph. * The paper is well written and easy to follow. Cons: * The model falls short in capturing the real system as it * The problem is considered from a learning perspective, whereas the quality of recommendation is overlooked. It is unclear how the learning can be used to improve performance under the proposed model. The authors do point out this shortcoming. * The authors claim that the experimental results validate the model. However, in the results section, the authors preprocess the dataset to remove users that have few edges. The authors should defend their validity claim when such alterations are made. * The definition of a match through the two directed edges seem cumbersome. The authors may consider a different description that can be generalized easily to non-binary ratings. * The proposed model is far from practical, such as binary preference. It is not clear whether the current insights will remain significant in such a setting. The author may consider commenting on that.